# The Antidepressant Effect of Resveratrol Is Related to Neuroplasticity Mediated by the ELAVL4-*Bdnf* mRNA Pathway

**DOI:** 10.3390/ijms26031113

**Published:** 2025-01-27

**Authors:** Hailong Ge, Lujia Si, Chen Li, Junjie Huang, Limin Sun, Lan Wu, Yinping Xie, Ling Xiao, Gaohua Wang

**Affiliations:** 1Department of Psychiatry, Renmin Hospital of Wuhan University, Wuhan 430060, China; hailongge@whu.edu.cn (H.G.); silj97@whu.edu.cn (L.S.); 2022283020071@whu.edu.cn (C.L.); hjj960606@whu.edu.cn (J.H.); wulan8@whu.edu.cn (L.W.); 2Department of Psychiatry, Institute of Neuropsychiatry, Renmin Hospital of Wuhan University, Wuhan 430060, China; rm004016@whu.edu.cn (L.S.); yinpingxie@whu.edu.cn (Y.X.); 3Taikang Center for Life and Medical Sciences, Wuhan University, Wuhan 430071, China

**Keywords:** depression, depressive-like behaviors, resveratrol, neuroplasticity, ELAVL4, *Bdnf* mRNA

## Abstract

Resveratrol, a plant-derived polyphenol, exhibits significant antidepressant effects and notably enhances neuroplasticity in neurological diseases. However, whether the antidepressant function of resveratrol is related to neuroplasticity remains uncertain, and the underlying mechanisms is poorly understood. This study aims to investigate the role and mechanism of resveratrol in neuroplasticity in depression. Here, we adopted the chronic unpredictable mild stress (CUMS) model and resveratrol intervention by oral gavage. Thereafter, behavioral tests confirmed resveratrol’s antidepressant effect, and Nissl staining, Golgi staining, and Western blotting (WB) were employed to assess the neuronal plasticity. Moreover, proteomic analysis and WB were used to screen and identify the key proteins. To investigate the downstream target of ELAV-like RNA-binding protein 4 (ELAVL4) (one of candidate genes), the RNA Interactome Database and the National Center for Biotechnology Information databases were utilized to predict the targets of ELAVL4. Finally, Quantitative PCR, WB, and Immunofluorescence were used to verify the prediction. Our results indicate that resveratrol alleviates CUMS-induced depressive-like behaviors accompanied by the restoration of impaired hippocampal neuroplasticity. Then, proteomic analysis shows that 351 differentially expressed proteins (DEPs) decrease after CUMS, while 24 DEPs increase remarkably with the resveratrol treatment. Among which, ELAVL4 is downregulated by CUMS, simultaneously increasing after resveratrol intervention, which acts as a protective protein in this process. Finally, brain-derived neurotrophic factor (*Bdnf*) mRNA is predicted to be the potential target of ELAVL4 and validated by molecular technologies. In conclusion, our findings demonstrate that resveratrol’s antidepressant efficacy is closely associated with ELAVL4, an RNA-binding protein, a mediated neuroplasticity pathway, potentially intersecting with the *Bdnf* mRNA. Overall, this research sheds light on the role of the ELAVL4-*Bdnf* mRNA pathway through neuroplasticity in resveratrol’s antidepressant action, which provides an mRNA regulation perspective for the development of novel antidepressants and understanding depression pathology.

## 1. Introduction

Depression is a serious mental disorder with a high prevalence and disability rate [1]. Over the past three decades, the global incidence of depression has increased by nearly 50%, affecting the physical and mental well-being and quality of life of more than 264 million individuals [2]. The main symptoms of depression are anhedonia, depressed mood, and cognitive and social impairment [3]. However, the current effectiveness of antidepressant medications is limited, patient compliance is low, and adverse effects persist as a major concern. In contrast, certain dietary components, such as resveratrol, tea polyphenols, and caffeine, demonstrate notable antidepressant properties with minimal side effects [4,5,6,7]. Furthermore, their consumption ensures better compliance compared to psychotropic drugs [8]. Hence, exploring the use of these dietary components as a viable adjunctive treatment for depression shows promise in enhancing the management of this mental disorder. Further research is essential to fully elucidate their potential benefits and to develop evidence-based recommendations for their integration into existing treatment regimens.

Resveratrol, one of the aforementioned components, known for its antidepressant property, is present in over 70 plant species, including grapes, berries, and peanuts. Among which, grapes and wine are the primary sources of resveratrol in our diet. Various studies have consistently demonstrated the remarkable antidepressant effect of resveratrol [9,10]. Nevertheless, although studies indicate that resveratrol can modulate neuroplasticity, the association between resveratrol’s antidepressant property and neuroplasticity remains unclear. For example, one study demonstrates that resveratrol enhances the plasticity of dopaminergic neurons, underscoring its potential role in modulating neuroplasticity [11]. Meanwhile, resveratrol has been shown to ameliorate cognitive impairments in rats through its neuroprotective effect [12]. Importantly, resveratrol has also been found to regulate long-term potentiation (LTP) in the hippocampus [13], a key electrophysiological phenomenon widely recognized as a cornerstone of neuroplasticity [14,15]. LTP provides a foundational framework for understanding the cellular mechanisms of learning and memory. It is characterized by a rapid and enduring increase in synaptic strength induced by the brief high-frequency stimulation of excitatory synapses. This sustained neuroplasticity enhancement, often persisting for days, plays a critical role in supporting cognitive functions and learning processes [14].

Neuroplasticity is a crucial hypothesis in understanding the pathological mechanisms underlying depression [16]. A recent study also proposes that synaptic plasticity is a core mechanism in the depression disorder, with other mechanisms affecting the disorder by intervening in neuroplasticity [17]. Various aspects of neuroplasticity regulation, including neurogenesis, neurotrophic factors, and synaptic function, collectively contribute to the modulation of personal emotion, cognition, and other significant functions [17,18]. In addition, within the realm of brain plasticity regulation, the hippocampus holds particular importance as it connects different emotion-related brain regions and nuclei including the prefrontal cortex, thalamus, amygdala, and nucleus accumbent [19]. Meanwhile, as reported, the hippocampal neuroplasticity is closely associated with depression due to its interaction with the neurogenesis, the hypothalamic–pituitary–adrenal axis, and other stress-induced mechanisms [20]. Therefore, we focused on hippocampal neuroplasticity as the central subject of our research into the antidepressant effect of resveratrol. Interestingly, neuroplasticity is regulated at multiple levels, including gene transcription, post-transcriptional modifications, protein expression, and post-translational modifications [21]. Although significant strides have been made in exploring the neuroplasticity hypothesis of depression on the levels of gene and protein expression and their modifications, studies addressing post-transcriptional modifications, particularly mRNA regulation, are still scarce.

ELAV-like RNA-binding protein 4 (ELAVL4), a highly conserved RNA-binding protein (RBP), contains three RNA-binding motifs that allow it to recognize and bind to mRNAs [22]. It plays a critical role in the stability and localization of mRNAs, providing a novel perspective on the neuroplasticity mechanisms in depression through the lens of mRNA regulation. In general, ELAVL4 is predominantly expressed in neurons and particularly abundant during the early stages of nervous system development, playing a crucial role in neurogenesis, nutritional factors, and synaptic function [22,23,24]. Studies reveal that several neurotrophins, such as the brain-derived neurotrophic factor (*Bdnf*) mRNA [25], nerve growth factor (*Ngf*) mRNA [26], growth-associated protein-43 (*Gap-43*) mRNA [27], and neuritin 1 (*Nrn1*) [28], are mainly involved in the neuroplasticity regulation of depression, all of which are the downstream mRNA targets of ELAVL4. Intriguingly, in the responses to antidepressant drugs, ELAVL4 is the gene with the highest percentage of single-nucleotide polymorphisms associated with positive outcomes [29]. These findings suggest that ELAVL4 may play a significant role in depression. Nevertheless, ELAVL4 and its concurrent downstream alterations remain elusive regarding its relationship to depression. Furthermore, the role of ELAVL4 in influencing hippocampal neuroplasticity in the antidepressant action of resveratrol is not yet fully understood.

As the last step, we explore the downstream molecular validating resveratrol’s antidepressant effect and measure alterations in hippocampal neuroplasticity following resveratrol treatment. Subsequently, hippocampal proteomic analysis identifies ELAVL4 as a key protein in the antidepressant action of resveratrol. Lastly, we predict *Bdnf* mRNA as the potential target of ELAVL4 and analyze changes in *Bdnf* mRNA and protein expression.

## 2. Results

### 2.1. Resveratrol Ameliorates Depressive-like Behaviors, Cognitive Impairment, and Social Deficits in the Chronic Unpredictable Mild Stress (CUMS) Rats

Before CUMS, there were no significant differences in the sucrose preference test (SPT) (Appendix A) and open field test (OFT) (Appendix A) among the four groups (*p* > 0.05). After CUMS, rats exhibited significant decreases in body weight (*p* < 0.0001, Figure 1B). The sucrose preference index in the SPT (Cums vs. Con) (*p* < 0.0001, Figure 1C) is declined by CUMS, along with prolonged immobility time (Cums vs. Con) (*p* < 0.0001, Figure 1D) and reduced swimming time (Cums vs. Con) (*p* < 0.001, Appendix A) in the forced swimming test (FST). The CUMS model significantly reduces total distance (*p* < 0.0001, Figure 1E), time in center (*p* < 0.0001, Figure 1F), and rearing frequency (*p* < 0.01, Figure 1G) in the OFT (Cums vs. Con). Notably, these behaviors are reversed by the administration of resveratrol (Res vs. Pbs) (*p* < 0.001 of sucrose preference index, *p* < 0.01 of immobility time, *p* < 0.01 of swimming time, *p* < 0.001 of total distance, *p* > 0.05 of time in center and *p* < 0.05 of frequency of rearing, Figure 1C–G and Appendix A). Resveratrol does not reverse the decrease in time in center (Res vs. Pbs), which may be consistent with previous findings suggesting that the time in center of OFT is more closely associated with anxiety-related behaviors. Overall, these findings validate the antidepressant and anxiolytic effects of resveratrol in mitigating CUMS-induced depressive-like and anxiety-like behaviors. In addition, to assess cognitive and social function, the novel object recognition test (NOR) and social interaction test (SIT) are conducted. Firstly, there were no significant differences in the M/N index among the four groups of rats during the second phase of NOR (*p* > 0.05, Appendix A). However, in the third phase, after CUMS exposure, a significant decline in the discrimination index was observed (Cums vs. Con) (*p* < 0.0001, Figure 1H). This decrease was markedly improved following resveratrol treatment (Res vs. Pbs) (*p* < 0.05, Figure 1H). In terms of social function, the rats show a significant decrease in the social index (Cums vs. Con) (*p* < 0.01, Figure 1I) and the social preference index (Cums vs. Con) (*p* < 0.05, Appendix A) after CUMS. However, resveratrol intervention effectively restores the sociability index (Res vs. Pbs) (*p* < 0.01, Figure 1I) without significant changes in the social preference index (Res vs. Pbs) (Appendix A). The findings of the NOR and SIT indicate that resveratrol may be beneficial for the cognition and social ability of rats.

### 2.2. Resveratrol Relieves the Hippocampal Neuroplasticity Damage Induced by CUMS

The Gene Ontology (GO) enrichment analysis of downregulated differentially expressed proteins (DEPs) (Cums vs. Con) (Figure 2A), upregulated DEPs (Cums vs. Con) (Figure 2B), downregulated DEPs (Res vs. Pbs) (Figure 2C), and upregulated DEPs (Res vs. Pbs) (Figure 2D) reveal significant changes in neuroplasticity-related biological processes and cellular components (For image details, please refer to Appendix A). These findings suggest that both CUMS intervention and resveratrol treatment have notable influences on neuroplasticity, highlighting its potential role in the therapeutic effects of resveratrol. To validate the predicted GO enrichment analysis findings, significant molecular experiments about neuroplasticity are conducted. Firstly, the expression of PSD95 shows a significant decrease after CUMS (Cums vs. Con) (*p* < 0.05, Figure 2E,F), which is increased by resveratrol intervention (Res vs. Pbs) (*p* < 0.05, Figure 2E,F). Then, the Nissl staining analysis demonstrates a notable reduction in positive neuron counts in both the hippocampal DG region (Cums vs. Con) (*p* < 0.01, Figure 2G–H) and CA3 region (Cums vs. Con) (*p* < 0.01, Figure 2G,I). After resveratrol intervention, there is a significant increase in the number of positive neurons in both the DG region (Res vs. Pbs) (*p* < 0.05, Figure 2G,H) and CA3 region (Res vs. Pbs) (*p* < 0.01, Figure 2G,I) of the hippocampus. Concurrently, in morphological characteristics, neurons in the DG and CA3 regions exhibit structural ambiguity, disorderly arrangements, and numerous vacuoles and pus-like features (Cums vs. Con) (Figure 2G). Conversely, neurons display neat and orderly arrangements, with most nuclei exhibiting an oval or round shape and increased cytoplasmic density (Res vs. Pbs) (Figure 2G). Furthermore, the Golgi staining assay similarly shows resveratrol’s improvement of impaired neuroplasticity by the CUMS model. The dendritic spine impairment after the CUMS model (Cums vs. Con) (*p* < 0.01, Figure 2J,K) is extremely restored by resveratrol (Res vs. Pbs) (*p* < 0.01, Figure 2J,K).

### 2.3. ELAVL4 Acts as a Pivotal Role in the Antidepressant Action of Resveratrol

Our analysis reveals that 351 downregulated and 514 upregulated DEPs are induced by CUMS. Meanwhile, resveratrol intervention results in 24 upregulated and 29 downregulated DEPs. To identify the key proteins involved in the antidepressant and neuroplasticity-enhancing actions of resveratrol, we intersect the downregulated DEPs (Cums vs. Con) with the upregulated DEPs (Res vs. Pbs), revealing one common protein (ELAVL4) (Figure 3A). Similarly, we intersect the upregulated DEPs (Cums vs. Con) with the downregulated DEPs (Res vs. Pbs), identifying four common proteins (trans-2,3-enoyl-CoA reductase (TECR), kininogen 2 (KNG2), kininogen 1 (KNG1), phosphodiesterase 1B (PDE1B)) (Figure 3B). Upon further review of the literature [28,30], we find that only ELAVL4 has potential associations with neuroplasticity. Accordingly, we select ELAVL4 as the focus on the role of resveratrol in neuroplasticity. Meanwhile, volcano plots highlight ELAVL4 as showing a more pronounced fold change compared to other proteins after CUMS, which is also significantly modified following resveratrol intervention (Figure 3C,D). To validate the proteomic results, Western blotting (WB) experiments are performed. The WB results demonstrate a significant decrease in ELAVL4 expression in the hippocampus after CUMS (Cums vs. Con) (*p* < 0.05, Figure 3E,F), which is markedly upregulated after resveratrol administration (Res vs. Pbs) (*p* < 0.05, Figure 3E,F). These findings suggest that ELAVL4 might play a pivotal role in antidepressant and neuroplasticity-enhancing properties of resveratrol.

To verify the association between the ELAVL4 levels in hippocampus with the rats’ depressive-like behaviors, simple linear regression analysis is used to reveal the correlations between ELAVL4 expression and behavioral parameters, which positively correlate with ELAVL4 expression. Specifically, the sucrose preference index shows a positive correlation with ELAVL4 expression (R^2^ = 0.5913, *p* = 0.0035, Figure 3G), while immobility time during forced swimming exhibits a negative correlation with the ELAVL4 expression (R^2^ = 0.5646, *p* = 0.0048, Figure 3H). In the OFT, the total distance moved (R^2^ = 0.4365, *p* = 0.0193, Figure 3I) and time in center (R^2^ = 0.4724, *p* = 0.0135, Figure 3J) display positive correlations with the ELAVL4 expression. Additionally, results from the NOR and SIT show positive correlations between the ELAVL4 expression and the discrimination index (R^2^ = 0.3677, *p* = 0.0366, Figure 3K) and the preference index (R^2^ = 0.3967, *p* = 0.0282, Figure 3L), respectively. These findings suggest that the cognition, memory, and social function impairments in rats are closely associated with the expression level of ELAVL4.

### 2.4. Bdnf Is the Potential Target Gene of ELAVL4

As a critical RBP, ELAVL4 plays an important role in neuroplasticity. We utilize the RNA Interactome Database (RNAInter) to identify its downstream target mRNAs, which shows two target mRNAs of ELAVL4 in rats: *Bdnf* mRNA and Beta-actin (*Actb*) mRNA. Concurrently, we search the NCBI database for proteins associated with neuroplasticity, resulting in 40 candidates. These two datasets yield a single overlap: *Bdnf* mRNA (Figure 4A). Meanwhile, RNAInter analysis confirms that *Bdnf* mRNA is a key downstream molecule of ELAVL4 in rats among the top 100 predicted interactions (Figure 4B,C). Figure 4D shows the structural binding between ELAVL4 and *Bdnf* mRNA visualized by AlphaFold 3.0. Furthermore, we examine the expression changes in *Bdnf* at the mRNA and protein levels. The results show that *Bdnf* mRNA and protein expression significantly decrease after CUMS (Cums vs. Con) (*p* < 0.01 for mRNA, *p* < 0.01 for protein, Figure 4E–G), while resveratrol intervention significantly increase their expression (Res vs. Pbs) (*p* < 0.05 for mRNA, *p* < 0.05 for protein, Figure 4E–G).

In addition, in order to investigate the potential interaction between ELAVL4 and BDNF, Immunofluorescence (IF) experiments are performed to analyze their expression and localization in the hippocampal DG and CA3 regions. The results demonstrate a significant downward trend in the expression of ELAVL4 in DG (Cums vs. Con) (*p* < 0.001, Figure 4H,I) and CA3 (Cums vs. Con) (*p* < 0.001, Figure 4K,L) regions. Consistent with the reduction in ELAVL4, BDNF expression also significantly decreased in the DG (Cums vs. Con) (*p* < 0.05, Figure 4H,J) and CA3 regions (Cums vs. Con) (*p* < 0.01, Figure 4K,M). Nevertheless, after resveratrol treatment, the decreasing trends of ELAVL4 and BDNF are reversed. The Res exhibits a significant increase in ELAVL4 in both the DG (Res vs. Pbs) (*p* < 0.01, Figure 4H,I) and CA3 (Res vs. Pbs) (*p* < 0.05, Figure 4K,L) regions, accompanied by an increase in BDNF expression in the DG (Res vs. Pbs) (*p* < 0.05, Figure 4H,J) and CA3 (Res vs. Pbs) (*p* < 0.01, Figure 4K,M) regions. Moreover, our results demonstrate that ELAVL4 is distributed in the nucleus and cytoplasm, consistent with its functional role. ELAVL4 may bind to *Bdnf* mRNA in the nucleus, reducing its degradation and facilitating directional movement to exert localization and other related functions. Conversely, BDNF does not enter the nucleus and is translated from mRNA outside the nucleus, subsequently playing a role in neurogenesis, differentiation, and repair. IF analysis reveals significant co-localization between ELAVL4 and BDNF in both the DG region (Pearson’s R = 0.63) and CA3 region (Pearson’s R = 0.68), indicating their close association (For image details, please refer to Appendix A).

## 3. Discussion

This study provides a comprehensive confirmation of the significant antidepressant and neuroplasticity-enhancing effects of resveratrol. Then, ELAVL4 in the hippocampus is identified as a key protein mediating resveratrol’s antidepressant action. Subsequently, RBP target mRNA analysis predicts *Bdnf* mRNA as a downstream target of ELAVL4 in rats. Furthermore, we validate the involvement of *Bdnf* in resveratrol-induced neuroplasticity restoration at both mRNA and protein levels. Overall, our findings suggest that the hippocampal ELAVL4-*Bdnf* mRNA pathway is a critical pathway through which resveratrol may restore neuroplasticity, thereby contributing to resveratrol’s antidepressant effect.

Despite the high efficacy, low side effects, and good compliance of resveratrol, the specific mechanisms underlying its antidepressant action remain unclear [31]. In this study, we reaffirm resveratrol’s antidepressant property, consistent with previous research [9,12]. Notably, neuroplasticity, as a central hypothesis in depression research, plays a pivotal role in understanding both the pathophysiology of depression and the mechanisms of antidepressant treatments [16]. Nevertheless, studies investigating the relationship between the antidepressant action of resveratrol with neuroplasticity is scarce. By CUMS and resveratrol intervention, our study successfully addresses this gap. We demonstrate that resveratrol significantly restores synaptic molecular gene expression reduced by CUMS and exerts a crucial protective role in neuroplasticity, thereby mediating its antidepressant effect. These findings align with studies on clinical antidepressants, as most of these drugs significantly enhance neuroplasticity [32,33]. For example, psychedelics promote neuroplasticity by directly binding to the BDNF receptor, tropomyosin receptor kinase B [34]. However, the specific proteins and pathways through which resveratrol regulates neuroplasticity remain unclear.

To explore potential mechanisms, we employ proteomic to identify key proteins involved in resveratrol’s antidepressant and neuroplasticity-enhancing effects. Given hippocampus’s critical role in depression [20] and its relevance to our previous studies, we focus on hippocampal proteomic. In the proteomic analysis, the overlap between the downregulated DEPs (Cums vs. Con) and the upregulated DEPs (Res vs. Pbs) is ELAVL4, and the overlap between the upregulated DEPs (Cums vs. Con) with the downregulated DEPs (Res vs. Pbs) is TECR, KNG1, KNG2, and PDE1B. Among the five proteins, only ELAVL4 is associated with neuroplasticity after a comprehensive literature review. In our study, ELAVL4 emerges as the neuroplasticity-related protein, prompting further investigation into its role in resveratrol-mediated neuroplasticity restoration and antidepressant property. This finding supports previous research demonstrating ELAVL4’s neuroprotective effects in nervous system diseases [35,36]. Moreover, our study extends this understanding to depression, highlighting ELAVL4 as a potential key protein in resveratrol’s antidepressant action on neuroplasticity.

Through RNAInter and NCBI database analyses, we identify *Bdnf* mRNA as a potential synaptic plasticity-related target of ELAVL4 in rats. Furthermore, we validate this protein-mRNA binding process at both the mRNA and protein levels in subsequent experiments. These findings are supported by studies in other species, such as Homo sapiens [26] and Mus musculus [37]. For instance, one study demonstrates that ELAVL4 regulates the stability and localization of *Bdnf* mRNA, further influencing neuroplasticity [26]. Taking everything into account, our discoveries align with previous studies, which reveal that *Bdnf* is essential for the antidepressant effect of resveratrol [38,39]. However, previous research predominantly concentrates on the transcriptional regulation and protein expression of BDNF with limited attention to its post-transcriptional modifications. Our study fills this gap by uncovering the regulation of *Bdnf* mRNA by the RNA-binding protein ELAVL4, which plays a critical role in resveratrol’s therapeutic effects.

It is still unclear whether other neurotrophic factors are involved in ELAVL4’s functions in antidepressant processes and the restoration of neuroplasticity. As an RBP, ELAVL4 stabilizes and processes the 3’ ends of numerous mRNAs, playing a pivotal role in regulating gene expression. Previous studies have identified ELAVL4 as an RBP targeting several neuroplasticity-related mRNAs, including *Bdnf* [25], *Ngf* [26], *Gap-43* [27], and *Nrn1* [28], all of which are closely associated with neuroplasticity and have well-established links to depression [40,41]. However, few research studies tend to explore the functions of these neurotrophic factor mRNAs in depression from the perspective of RBPs. This study addresses this limitation using Quantitative PCR (qPCR), which reveals that, among the key depression-associated neurotrophic factors, only *Bdnf* mRNA exhibits significant changes (Appendix A). These findings further indicate the critical role of the ELAVL4-*Bdnf* mRNA pathway in mediating resveratrol’s antidepressant effect and its contribution to the restoration of neuroplasticity. Furthermore, this study underlines the necessity for deeper investigation into mRNA-protein regulatory mechanisms within the context of depression research.

Finally, WikiPathways enrichment analysis (WPEA) reveals that the common pathway in downregulation and upregulation under Cums/Con (Appendix A) and upregulation and downregulation under Res/Pbs (Appendix A) is the beta-oxidation of unsaturated fatty acids, suggesting a potential role of metabolic pathways in resveratrol’s antidepressant effect. Similar findings are observed in the Reactome enrichment analysis (Appendix A), further supporting the involvement of metabolism in resveratrol’s therapeutic property. These results are consistent with previous studies indicating that the antidepressant property of resveratrol may be linked to metabolic pathways [42]. However, previous research has not considered neuroplasticity, which is highlighted in our study. We propose that the promotion of neuroplasticity by resveratrol and changes in metabolism may occur in a time-coordinated manner, suggesting a potential interaction between the two. This makes sense, as neuroplasticity requires significant energy, and resveratrol’s action on neuroplasticity may be related to its regulation of metabolism. However, further research should deeply explore the interplay between resveratrol’s neuroplasticity-promoting effect and its metabolic regulation, which is a constraint of this study.

This study also has limitations that warrant acknowledgment. First, the water deprivation involved in the CUMS protocol, combined with the daily administration of approximately 1 mL of water via gavage in the Pbs group (as the control for the Res group), may have partially influenced the outcomes of the OFT and NOR. While this represents a significant factor, it is not the sole contributor. Future studies will address this limitation by utilizing alternative depression animal models or implementing refined experimental designs to minimize such confounding effects. Second, there remains uncertainty regarding whether the concentration of resveratrol used in this study accurately reflects the dosage typically consumed in clinical settings, given differences in digestive efficiency and metabolic systems between rats and humans. Further clinical research is needed to investigate the efficacy of resveratrol at varying doses in patients with depression. In addition, there are ongoing debates about whether the metabolism of resveratrol in rats is consistent with those in humans, due to substantial differences in their metabolic systems. Future studies should conduct comparative analyses to explore these metabolic variations to address the limitations of this research.

## 4. Materials and Methods

### 4.1. Animals

Male Sprague Dawley (SD) rats (140–160 g, 6 weeks old, N = 40) were obtained from China Three Gorges University. All rats were housed in standard laboratory conditions at the Center for Experimental Animals at Wuhan University in China and diets and water were supplied ad libitum. The feeding environment was maintained at a temperature of 24 ± 2 °C, a relative humidity of 65% ± 5%, and a 12-h light/dark cycle, with the light phase set from 8:00 to 20:00. The rats were housed in cages measuring 55 cm × 40 cm × 20 cm, with five rats per cage. All experimental animals were approved by the Institutional Animal Care Committee of Renmin Hospital, Wuhan University, and experiments were performed in accordance with the guidelines. Additionally, these animal experiments were conducted in full compliance with the Animal Research: Reporting of In Vivo Experiments (ARRIVE) guidelines.

### 4.2. Experimental Grouping and Design

After 7 days of adaptation and 4 days of baseline behaviors tests, all rats were randomly divided into four groups (N = 10, respectively): the control group (Con), the chronic unpredictable mild stress group (Cums), the phosphate-buffered saline group (Pbs), and the resveratrol group (Res). In the Con, rats had unrestricted access to food and water without any additional stimulation (Figure 1A). The Cums, Pbs, and Res were subjected to a 28-day CUMS model (the specific process is described below). During the entire CUMS process, the Pbs received phosphate-buffered saline containing 5% DMSO intervention, while the Res received resveratrol intervention, both via oral gavage. At the end of the 28-day period, all rats underwent behavioral tests. Then, samples were collected for further analysis (Figure 1A).

### 4.3. Stressors and Operational Processes of CUMS and Drug Intervention

The stress protocol consisted of a randomized series of mild stressors: (A) tilting the cage at 45° for 24 h; (B) damp bedding for 24 h; (C) 24-h food deprivation; (D) 24-h water deprivation; (E) subjecting to restraint stress for 2 h; (F) briefly clamping the tail for 5 min; (G) forced swimming in water at 4 °C for 5 min; and (H) forced swimming in water at 45 °C for 5 min. Rats in the CUMS model were exposed to two stressors. The same type of stressor cannot be used two days in a row. Resveratrol (R0071, TCI, Shanghai, China), dissolved in phosphate-buffered saline containing 5% DMSO, was administered by gavage (80 mg/kg).

### 4.4. Behavioral Testing

#### 4.4.1. Body Weight (BW)

BW data were collected every Saturday at 8:30.

#### 4.4.2. SPT

On Day 1, all rats were housed individually, and two bottles of 1% sucrose solution were placed in each cage for adaptation. On Day 2, one of the bottles was replaced with tap water for the following 12 h, following 12 h of water and food deprivation. On Day 3, rats were presented with two bottles simultaneously, and their intake of one day was measured: 1% sucrose consumption (M), and tap water consumption (N). The sucrose preference was calculated as [M/(M + N)] × 100%.

#### 4.4.3. OFT

The OFT started at 8:30. In this study, a simple and spacious open field measuring 100 cm × 100 cm × 30cm was employed. To accurately capture their behaviors, we utilized the Ethovision XT 11.5 tracking system. Each rat was placed in the middle of the open field to explore spontaneously for 5 min. The Ethovision XT 11.5 tracking system was used to capture their behaviors such as the total distance traveled by the rats, the time spent in the central arena (measuring 33.3 cm × 33.3 cm), and the frequency of rearing behaviors during the observation period.

#### 4.4.4. FST

The FST commenced at 8:30. Each rat was subjected to forced swimming in a transparent glass cylinder (height of 40 cm, diameter of 28 cm, water depth of 30 cm, and temperature of 25 ± 1 °C). All rats were put in the cylinders of 6 min, and the immobility time during the 4 min before the end was recorded. This test employed a blinding protocol, ensuring experimenters were uninformed about the treatment groups.

#### 4.4.5. NOR

The NOR was conducted to evaluate the memory capacity of rodents (Appendix A). In brief, the experiment consisted of three phases. During the first phase (adaptation) on Day 1, rats were given a 5 min acclimation period in a 100 cm × 100 cm open field. The second phase, conducted on Day 2, involved exposing the rats to two identical red cylinders (10 cm in diameter, 20 cm in height) for 10 min. The time the rats spent within 1 cm of each cylinder was recorded, with the time spent near cylinder 1 denoted as M and the time near cylinder 2 as N. The discrimination index for this phase, calculated as [M/(M + N)] × 100%, was used to assess potential biases caused by the experimental setup. The third phase, conducted one hour after the second phase on Day 2, replaced one red cylinder with a green cube (15 cm × 15 cm × 15 cm). Rats were then allowed to explore the objects in the open field for 5 min. The time spent within 1 cm of the green cube and the remaining red cylinder was recorded as X and Y, respectively. The recognition index, calculated as [X/(X + Y)] × 100%, was used as a measure of recognition memory. Given the innate preference of rats for novel objects, they tend to spend more time exploring the new object. Consequently, a decline in cognitive or memory function would result in a reduced recognition index. The Ethovision XT 11.5 tracking system was utilized to record data.

#### 4.4.6. SIT

The SIT commenced at 8:30 (Appendix A). In this test, the experimental apparatus was composed of three rectangular boxes, each with internal dimensions of 40 cm × 60 cm × 40 cm. The boxes were separated by transparent resin-glass dividers, and there are channels of 10 cm × 20 cm in the middle that allow for communication between the three boxes. Each of the left and right parts had a cage holding a strange rat. In addition, the test consisted of three phases. The first phase, adaptation, lasted for 10 min, during which the test rat was placed in the middle of the area to acclimate to the environment. In the second phase, a novel SD rat (T1) of the same sex, age, and weight was placed in one wire cage, while the opposite side remained empty. The test rat was allowed to explore all chambers freely for 10 min. During this phase, the time that the test rat spent within 1 cm of the cage containing T1 was recorded as T1a, and the time spent within 1 cm of the empty cage was recorded as T0. The sociability index was calculated using the following formula: [T1a/(T1a + T0)] × 100%. In the third phase, another novel SD rat (T2), matched for sex, age, and weight, was placed in a separate wire cage. During this 10 min phase, the time spent within 1 cm of the cage containing T1 was recorded as T1b, and the time spent within 1 cm of the cage containing T2 was recorded as T2a. The social preference index was calculated using the following formula: [T2a/(T1b + T2a)] × 100%. The duration of contact with T1 and T2 during the sociability and social preference phases was automatically recorded using SuperMaze V2.0 software.

### 4.5. Proteomic Assays

Quantitative proteomic analysis (iProteome, Shanghai, China) was employed to precisely identify and quantify the protein expression profile in rat’s hippocampus. This approach allowed to screen and identify DEPs following CUMS modeling and resveratrol intervention. Furthermore, bioinformatic analysis involving GO, WPEA, Reactome enrichment analysis revealed alterations in the pathological and physiological processes of the hippocampal region.

### 4.6. Analysis of the Potential Neuroplasticity-Related Target mRNAs of ELAVL4

RNAInter, which is freely accessible at http://www.rnainter.org (accessed on 16 November 2024), was initially analyzed using the keywords “ELAVL4” and “rattus norvegicus”, and the corresponding data were extracted. The National Center for Biotechnology Information (NCBI) database at https://www.ncbi.nlm.nih.gov/ (accessed on 16 November 2024) was queried using the keywords “neuroplasticity” and “rattus norvegicus” to identify neuroplasticity relative genes in rats. The intersection of these two datasets was analyzed, predicting that BDNF mRNA might be the target of ELAVL4 associated with neuroplasticity.

### 4.7. Structure Prediction of the Interaction Between ELAVL4 and BDNF mRNA

Complex structure prediction was conducted with AlphaFold3 (DeepMind, London, UK), and the ELAVL4 and BDNF mRNA sequences were obtained in the NCBI database at https://www.ncbi.nlm.nih.gov/ (accessed on 18 November 2024).

### 4.8. WB

The total proteins of hippocampal tissue were extracted with a high RIPA buffer (Solarbio, Beijing, China). Subsequently, resulting proteins (25 µg) were separated by 10% SDS polyacrylamide gel electrophoresis. The proteins were then transferred onto polyvinylidene difluoride membranes (pore size: 0.45 μm, Merck Millipore, Darmstadt, Ireland). After blocking with fast-blocking Western (PS108P, Epizyme, Shanghai, China), the membranes were incubated overnight with primary mouse anti-ELAVL4 (dilution 1:1000, 67835-1-lg, Proteintech, Chicago, IL, USA), rabbit anti-BDNF antibody (dilution 1:1000, ER130915, Huabio, Hangzhou, China), rabbit anti-PSD95 antibody (dilution 1:1000, ET1602-20, Huabio, Hangzhou, China) and mouse anti-GAPDH antibody (dilution 1:5000, 60004-1-Ig, Proteintech, Chicago, IL, USA) antibodies overnight at 4 °C. After being washed five times with 0.1% TBST, the membranes were incubated with the HRP-Goat Anti-Mouse Recombinant Secondary Antibody (dilution 1:5000, RGAM001, Proteintech, Chicago, IL, USA) and HRP-Goat Anti-Rabbit Recombinant Secondary Antibody (dilution 1:5000, RGAR001, Proteintech, Chicago, IL, USA) at 25 °C. Signal visualization was achieved using Chemiluminescence Solution P1000 (P1000-25, Applygen, Beijing, China) and the ChemiDocTouch Imaging System (Bio-Rad, Hercules, CA, USA), and signal intensity was quantified using Bio-Rad Laboratories’ Image Lab software 5.2.

### 4.9. qPCR

The total RNA of the hippocampus was extracted using the Triquick Reagent (Solarbio, Beijing, China), and the first strand cDNA synthesized by kits (K1621, Thermo Scientific, Waltham, MA, USA). The qPCR was performed with corresponding primers (listed in Table 1) and the mixes (172-5122, Bio-Rad, Hercules, CA, USA). Data were analyzed by the 2^−ΔΔCt^ method.

### 4.10. IF

Brain sections were fixed by Servicebio (Wuhan, China) and slices of the rats’ hippocampus measuring 5 μm in length were utilized. Then, the sections were treated with dewaxing and rehydration processes. And the sections were permeabilized using 0.5% Triton X-100 (Biosharp, Anhui, China) for 20 min. The samples were then incubated in 5% BSA (4240GR100, BioFroxx, Bavaria, Germany) for 1 h and subsequently treated with primary mouse anti-ELAVL4 antibody (dilution 1:1000, 67835-1-lg, Proteintech, Chicago, IL, USA), and rabbit anti-BDNF antibody (dilution 1:1000, ER130915, Huabio, Hangzhou, China) overnight at 4 °C. After secondary antibodies (Multi-rAb CoraLite^®^ Plus 594-Goat Anti-Rabbit Recombinant Secondary Antibody (RGAR004, Proteintech, Wuhan, China), Multi-rAb CoraLite^®^ Plus 488-Goat Anti-Mouse Recombinant Secondary Antibody (H + L) (RGAM002, Proteintech, China)) were incubated for 1 h in the absence of light, fluorescence was assessed using a Leica fluorescence microscope (TCSSP8, Leica, Wetzlar, Germany). ImageJ software 1.54m (National Institutes of Health, Bethesda, MD, USA) was used to analyze the IF results.

### 4.11. Nissl Staining and Golgi Staining

For Nissl staining, the sections were treated with dewaxing and rehydration processes. Then, they were stained with Nissl dye (Solarbio Life Science, G1430, Beijing, China) at 56 °C for 1 h. After staining, the sections were washed and differentiated in 95% alcohol. Finally, the average number of Nissl-positive cells were counted for analysis. In addition, Golgi staining was sent by Servicebio (Beijing, China).

### 4.12. Statistical Analysis

SPSS version 25.0 (IBM Corporation, New York, NY, USA) was used for statistical analysis. The Shapiro-Wilkes normality test and Levene’s test were applied to evaluate variance distribution and homogeneity, respectively. The one-way and two-way ANOVA followed by Tukey’s post hoc test or Friedman test (for non-Gaussian distributions) was performed among the four groups (Con, Cums, Pbs, and Res). Statistical significance was set at *p* < 0.05. Graphs were generated using GraphPad Prism version 9.0.0 (GraphPad, San Diego, CA, USA). In addition, all statistical data are presented in Table 2.

## 5. Conclusions

In summary, this study demonstrates that the ELAVL4-BDNF mRNA pathway in the hippocampus plays a pivotal role in mediating neuroplasticity, involved in the antidepressant effect of resveratrol. These findings highlight a novel pathway underlying resveratrol’s antidepressant property, offering an mRNA regulation perspective for the development of new antidepressant therapies and advancing our understanding of the pathological mechanisms of depression.

## Figures and Tables

**Figure 1 ijms-26-01113-f001:**
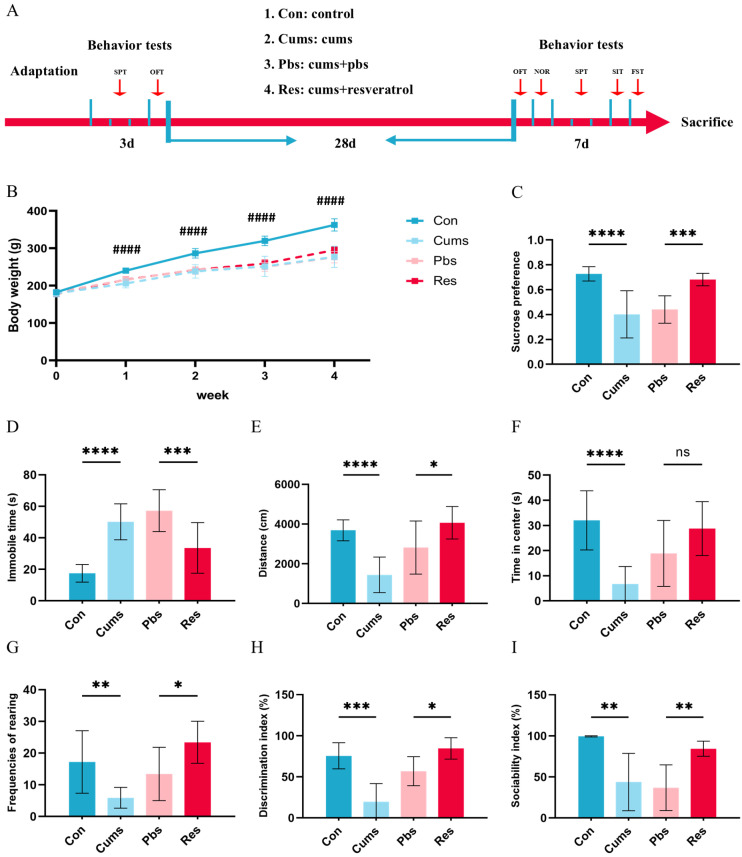
Resveratrol ameliorated depressive-like behaviors in rats: (**A**) Experimental design and timeline. (**B**) Body weight across groups (N = 10). (**C**) Sucrose preference index in sucrose preference test (SPT) (N = 10). (**D**) Immobility time in forced swimming test (FST) (N = 10). (**E**) Total distance of open field test (OFT) (N = 10). (**F**) Time in center of OFT (N = 10). (**G**) Frequencies of rearing in OFT (N = 10). (**H**) Discrimination index of novel object recognition test (NOR) (N = 6). (**I**) Sociability index of social interaction test (SIT) (N = 6). Statistical analyses are conducted using one-way ANOVA with Tukey’s post hoc test (#### *p* < 0.0001 (Cums vs. Con), ns: no statistical significance, * *p* < 0.05, ** *p* < 0.01, *** *p* < 0.001, and **** *p* < 0.0001).

**Figure 2 ijms-26-01113-f002:**
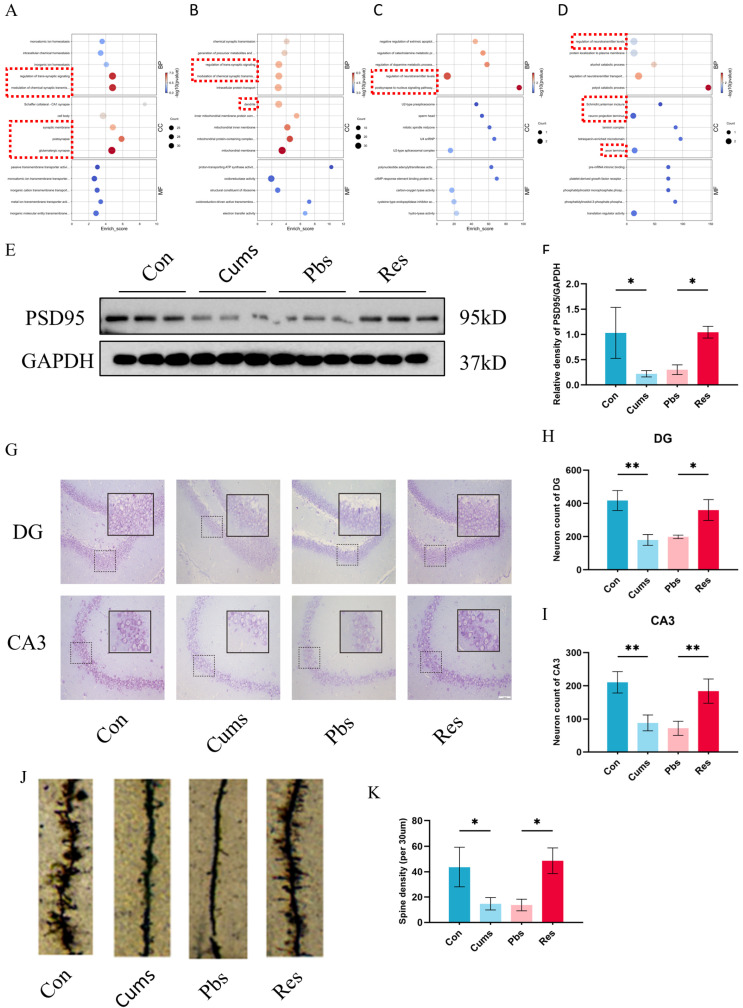
Resveratrol enhanced hippocampal neuroplasticity in rats: (**A**–**D**) Gene Ontology (GO) enrichment analysis of downregulated DEPs (Cums vs. Con) (**A**), upregulated DEPs (Cums vs. Con) (**B**), downregulated DEPs (Res vs. Pbs) (**C**), and upregulated DEPs (Res vs. Pbs) (**D**). (**E**,**F**) Relative expression of PSD95 normalized to GAPDH (N = 3). (**G**–**I**) Positive neuron counts in the DG and CA3 regions based on Nissl staining. Scale bars represent 75 μm. (**J**,**K**) Representative dendritic spine images and spine density measurements across the four groups (N = 3). Scale bar represents 5 μm. The red dotted box highlights changes associated with neuroplasticity. Statistical analyses are conducted using one-way ANOVA with Tukey’s post hoc test (* *p* < 0.05, ** *p* < 0.01.)

**Figure 3 ijms-26-01113-f003:**
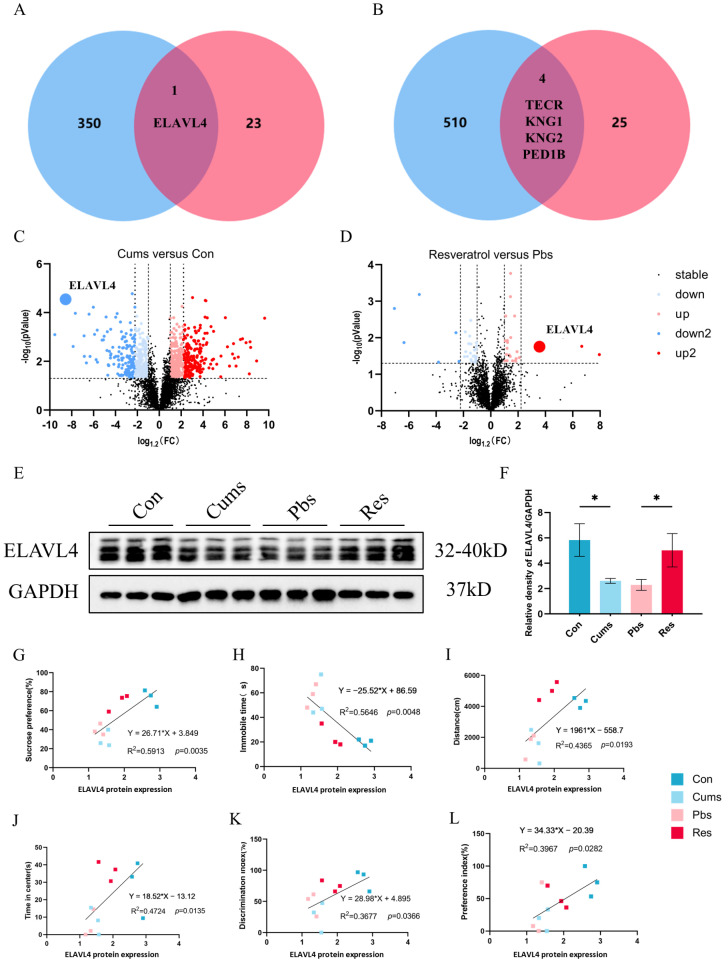
ELAVL4 as a key protein in the antidepressant effect of resveratrol: (**A**) Venn diagram of hippocampal proteomic showing downregulated DEPs (Cums vs. Con) and upregulated DEPs (Res vs. Pbs). (**B**) Venn diagram of hippocampal proteomic showing upregulated DEPs (Cums vs. Con) and downregulated DEPs (Res vs. Pbs). (**C**,**D**) Volcano plots of ELAVL4 expression in hippocampal proteomics, highlighting upregulated and downregulated proteins (FC > 1.2) and significantly altered proteins (FC > 1.5). (**E**,**F**) Relative expression of ELAVL4 normalized to GAPDH in the four groups (N = 3). The linear correlations between ELAVL4 expression and the sucrose preference index in the SPT (**G**), immobile time in FST (**H**), total distance of OFT (**I**), time in center of OFT (**J**), discrimination index of NOR (**K**), preference index of SIT (**L**). Statistical analyses are performed using one-way ANOVA with Tukey’s post hoc test (* *p* < 0.05).

**Figure 4 ijms-26-01113-f004:**
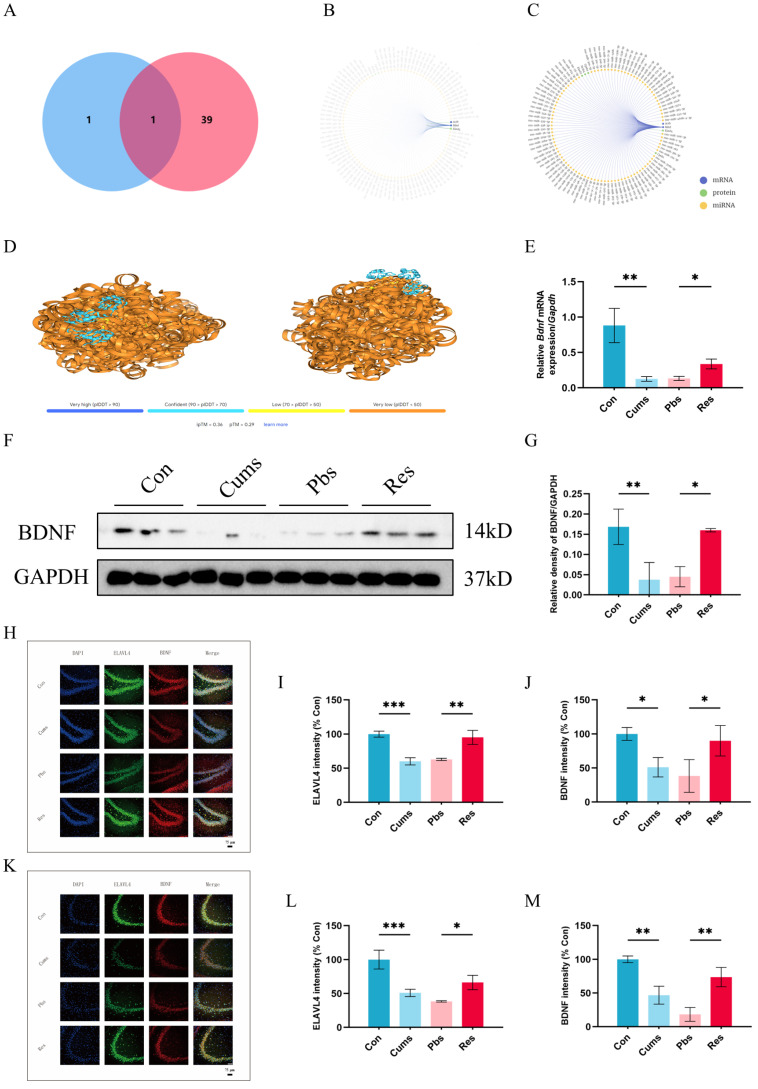
*Bdnf* as a potential downstream target of ELAVL4: (**A**) Venn diagram showing the overlap between ELAVL4 target mRNAs and neuroplasticity-related genes. (**B**,**C**) Interaction network of the top 100 interactions of ELAVL4 or *Bdnf* mRNA. (**D**) AlphaFold3-predicted structural interaction between ELAVL4 protein and *Bdnf* mRNA. (**E**) Relative *Bdnf* expression/Gapdh mRNA expression among the four groups. (**F**,**G**) Relative expression of BDNF normalized to GAPDH (N = 3). (**H**–**M**) Representative immunofluorescence images showing ELAVL4 and BDNF expression in the DG and CA3 region of the hippocampus (N = 3). Scale bars represent 75 μm. Statistical analyses are performed using one-way ANOVA with Tukey’s post hoc test. (* *p* < 0.05, ** *p* < 0.01, and *** *p* < 0.001).

**Table 1 ijms-26-01113-t001:** Primers for qPCR.

Gene	Forward Primer	Reverse Primer
NRN1	tcttacggattgccaggaag	gctaaagctgccgagagaga
GAP43	ggctctgctactaccgatgc	gacggcgagttatcagtggt
BDNF	tacctggatgccgcaaacat	tggccttttgataccgggac
NGF	acctcttcggacactctgga	gtccgtggctgtggtcttat
GAPDH	tgatgggtgtgaaccacgag	agtgatggcatggactgtgg

**Table 2 ijms-26-01113-t002:** All statistical data.

	F (DFn, DFd)	*p* Value
Figure 1C	F (3, 36) = 20.38	*p* < 0.0001
Figure 1D	F (3, 36) = 21.19	*p* < 0.0001
Figure 1E	F (3, 36) = 15.35	*p* < 0.0001
Figure 1F	F (3, 36) = 10.92	*p* < 0.0001
Figure 1G	F (3, 36) = 9.649	*p* < 0.0001
Figure 1H	F (3, 20) = 16.24	*p* < 0.0001
Figure 1I	F (3, 20) = 10.76	*p* = 0.0002
Figure 2F	F (3, 8) = 18.52	*p* = 0.0006
Figure 2H	F (3, 8) = 19.07	*p* = 0.0005
Figure 2I	F (3, 8) = 16.70	*p* = 0.0008
Figure 2K	F (3, 8) = 10.65	*p* = 0.0036
Figure 3F	F (3, 8) = 23.59	*p* = 0.0003
Figure 4E	F (3, 20) = 46.37	*p* < 0.0001
Figure 4G	F (3, 8) = 35.77	*p* < 0.0001
Figure 4I	F (3, 8) = 33.59	*p* < 0.0001
Figure 4J	F (3, 8) = 7.780	*p* = 0.0093
Figure 4L	F (3, 8) = 25.38	*p* = 0.0002
Figure 4M	F (3, 8) = 28.92	*p* = 0.0001
Appendix A	F (3, 36) = 1.820	*p* = 0.1609
Appendix A	F (3, 36) = 0.9385	*p* = 0.4322
Appendix A	F (3, 36) = 1.056	*p* = 0.3800
Appendix A	F (3, 36) = 0.4159	*p* = 0.7426
Appendix A	F (3, 20) = 7.547	*p* = 0.0014
Appendix A	F (3, 20) = 15.76	*p* < 0.0001
Appendix A (groups)	F (1.748, 8.742) = 117.8	*p* < 0.0001
Appendix A (genes)	F (1.250, 6.250) = 12.28	*p* = 0.0100

## Data Availability

The data will be available upon request from the authors.

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
