# Peer review of "The Antidepressant Effect of Resveratrol Is Related to Neuroplasticity Mediated by the ELAVL4-*Bdnf* mRNA Pathway"

_ijms, 2025, doi:10.3390/ijms26031113_

Round 1
Reviewer 1 Report
Comments and Suggestions for Authors
In this study, Ge et al. observed that in the Chronic Unpredictable Mild Stress (CUMS) model, the expression levels of ELAVL4 and BDNF were reduced. However, treatment with resveratrol alleviated depression-like behaviors and increased the levels of both ELAVL4 and BDNF.
Figure 1
1.1 How are the social index and social preference index calculated? In Figure 1I, the social index in the control group is nearly 100%. Does this indicate that the mice do not interact with the empty chamber?
1.2 In the Open Field Test (OFT) and Novel Object Recognition (NOR) test, the PBS treatment increased parameters such as distance traveled, time spent in the center, frequency of rearing, and the discrimination index. This result seems unreasonable and requires further explanation.
Figure 2
In Figures 2A-D, the font size is too small to be clearly read.
Figure 3
The two parts of Figure 3 and their accompanying legend should be combined for clarity.
Figure 4
The three parts of Figure 4 and their accompanying legend should also be combined for clarity.
Author Response
Dear professor,
I hope this message finds you well. I would like to sincerely thank you for your valuable suggestions on our manuscript. Your feedback has greatly enhanced the quality, rigor, and clarity of our study. Based on your recommendations, we have now completed the necessary revisions, and the revised manuscript along with our responses (Author's response to reviewer 1) is being submitted through the system.
Please feel free to review the updates at your convenience. Once again, thank you for your guidance and support throughout this process. Wishing you all the best and continued success in your work.
Warm regards,
Authors of this manuscript
Wuhan university
January 17, 2025

Reviewer 2 Report
Comments and Suggestions for Authors
The manuscript by Ge and colleagues is the result of a huge amount of work, is well-written and data are of good originality. In spite of it some issues require a careful consideration.
Major concerns
Introduction, lines 58-65.
Authors should improve the rational underlying the present study. The relative sentences are contradictory. Please also specify that long-term potentiation is the electrophysiological correlate of memory.
2. Results
NORT test
Total exploration times of objects during the sample T1 and the choice trial T2 should be reported. This parameter indicates whether or not unspecific factors (sensorimotor, attentional, motivational) had confounded animals' cognitive performance.
Did an anxiolytic effect of resveratrol was revealed in the open field test? Did rats' grooming activity was recorded in the open field test?
Please specify how was selected the dose of resveratrol used in the present study.
Minor concerns
line 47. Please report some of the dietary components.
line 127. "cognition and memory function" have identical significance.
Appropriate references or all tests are missed.
Please report the size of the cages and the number of rats per cage.
Line 339. NORT evaluates recognition memory.
Based on which evidence the duration of T1, T2 and that the intertrial interval were chosen?
Author Response
Dear professor,
I hope this message finds you well. I would like to sincerely thank you for your valuable suggestions on our manuscript. Your feedback has greatly enhanced the quality, rigor, and clarity of our study. Based on your recommendations, we have now completed the necessary revisions, and the revised manuscript along with our responses (Author's response to reviewer 2) is being submitted through the system.
Please feel free to review the updates at your convenience. Once again, thank you for your guidance and support throughout this process. Wishing you all the best and continued success in your work.
Warm regards,
Authors of this manuscript
Wuhan university
January 17, 2025

Round 2
Reviewer 2 Report
Comments and Suggestions for Authors
Manuscript has substantially been improved respect to its former version. It can now be considered for publication.